# Acute and Chronic Resistance Training, Acute Endurance Exercise, nor Physiologically Plausible Lactate In Vitro Affect Skeletal Muscle Lactylation

**DOI:** 10.3390/ijms252212216

**Published:** 2024-11-14

**Authors:** Madison L. Mattingly, Derick A. Anglin, Bradley A. Ruple, Maira C. Scarpelli, Joao G. Bergamasco, Joshua S. Godwin, Christopher B. Mobley, Andrew D. Frugé, Cleiton A. Libardi, Michael D. Roberts

**Affiliations:** 1School of Kinesiology, Auburn University, Auburn, AL 36849, USA; 2Geriatric Research Education and Clinical Center, George E. Whalen VA Medical Center, Salt Lake City, UT 84148, USA; 3Department of Internal Medicine, Division of Geriatrics, University of Utah, Salt Lake City, UT 84112, USA; 4MUSCULAB—Laboratory of Neuromuscular Adaptations to Resistance Training, Department of Physical Education, Federal University of São Carlos—UFSCar, São Carlos 13565-905, Brazil; maira@ufscar.br (M.C.S.);; 5College of Nursing, Auburn University, Auburn, AL 36849, USA

**Keywords:** skeletal muscle, hypertrophy, protein acetylation, protein lactylation

## Abstract

We examined changes in skeletal muscle protein lactylation and acetylation in response to acute resistance exercise, chronic resistance training (RT), and a single endurance cycling bout. Additionally, we performed in vitro experiments to determine if different sodium lactate treatments affect myotube protein lactylation and acetylation. The acute and chronic RT study (12 college-aged participants) consisted of 10 weeks of unilateral leg extensor RT with vastus lateralis (VL) biopsies taken at baseline, 24 h following the first RT bout, and the morning of the last day of the RT bout. For the acute cycling study (9 college-aged participants), VL biopsies were obtained before, 2 h after, and 8 h after 60 min of cycling. For in vitro experiments, C2C12 myotubes were treated with varying levels of sodium lactate, including LOW (1 mM for 24 h), HIGH (10 mM for 24 h), and PULSE (10 mM for 30 min followed by 1 mM for 23.5-h). Neither acute nor chronic RT significantly affected nuclear or cytoplasmic protein lactylation. However, cytoplasmic protein acetylation was significantly reduced following one RT bout (−15%, *p* = 0.002) and chronic RT (−16%, *p* = 0.006). Cycling did not acutely alter post-exercise global protein lactylation or acetylation patterns. Lastly, varying 24 h lactate treatments did not alter nuclear or cytoplasmic protein lactylation or acetylation, cytoplasmic protein synthesis levels, or myotube diameters. These findings continue to support the idea that exercise induces more dynamic changes in skeletal muscle protein acetylation, but not lactylation. However, further human research with more sampling timepoints and a lactylomics approach are needed to determine if, at all, different exercise modalities affect skeletal muscle protein lactylation.

## 1. Introduction

Adaptive responses to resistance training include increases in strength, skeletal muscle hypertrophy, and metabolic resilience [1,2]. One mechanism by which this can occur is through gene regulation via epigenetic modifications to DNA (i.e., CpG site methylation) and histones [3,4]. Histone acetylation is a well-documented post-translational and epigenetic modification that occurs with different modalities of exercise training [5]. More recent in vitro evidence additionally supports that histones can be lactylated in response to cellular stress [6], and this acts competitively with acetylation given that both forms of post-translational modification occur on lysine residues. Since this discovery, lactylation has been found to be linked with numerous processes in tissues and cells, including cardiac regulation, macrophage function, osteoblast cell differentiation, sepsis and cancer, and myogenesis [7,8,9,10,11].

While examining the histone and/or global protein lactylation responses to exercise stimuli may provide fresh perspectives for understanding exercise gene/protein regulation, it has been less thoroughly clarified relative to acetylation. In this regard, histone acetylation has been shown to be altered with acute and chronic resistance training as well as aerobic training [12,13]. Moreover, skeletal muscle transcription factors and metabolic enzymes can exhibit altered acetylation states, thereby affecting their functions [14]. Conversely, only one rodent study to date has reported that treadmill exercise transiently increases skeletal muscle protein lactylation [15]. Contrary to these data are human data from our laboratory indicating that acute resistance exercise, which leads to robust increases in peri-exercise blood lactate levels, did not affect 3 h or 6 h post-exercise skeletal muscle protein lactylation levels and/or the mRNA expression of associated genes [16]. However, aside from these two studies providing equivocal data (potentially due to species and exercise modality differences), there is an overall lack of human evidence as to whether different exercise modalities can affect skeletal muscle protein lactylation. Likewise, it is unclear whether dynamic alterations in post-exercise protein acetylation levels are inversely related to protein lactylation levels, as reported by Zhang et al. in vitro [6].

Therefore, one aim of this study was to determine if one bout or chronic resistance training and/or one bout of endurance exercise affected various markers of skeletal muscle protein acetylation and lactylation in humans. Moreover, we sought to determine how either pulse or 24 h sodium lactate treatments affected similar outcomes in C2C12 myotubes (See Figure 1). Due to our previous null findings, we hypothesized that markers associated with protein lactylation would either remain unaltered or decrease in response to different exercise paradigms. Moreover, in line with prior literature, we hypothesized that markers associated with protein acetylation would be dynamically altered. Lastly, we hypothesized that lactate administration would not alter markers associated with protein lactylation in vitro.

## 2. Results

The results for the acute and chronic resistance training effects on muscle protein acetylation and lactylation markers are presented in Figure 2. Cytoplasmic and nuclear protein lactylation showed no significant model effects (*p* = 0.879 and *p* = 0.154, Figure 2a,b). Although nuclear protein acetylation showed no significant model effect (*p* = 0.109, Figure 2d), cytoplasmic protein acetylation demonstrated a significant model effect (*p* < 0.001, Figure 2c) and was significantly lower at 24 h after bout 1 and the intervention compared to before (*p* = 0.002 and *p* = 0.006, respectively). Lastly, nuclear H3K9 acetylation showed no significant model effect (*p* = 0.296, Figure 2e).

### 2.1. One Bout of Resistance Exercise, but Not Chronic Training, Significantly Decreases Select Enzymes Involved in Protein Acetylation and Lactylation

The results for the effects of acute and chronic resistance training on select enzymes involved with protein acetylation (HDAC2/6) and lactylation (LDHA) are presented in Figure 3. Cytoplasmic and nuclear HDAC2 protein expression showed no significant model effects (*p* = 0.755 and *p* = 0.393, respectively, Figure 3a,b). However, cytoplasmic and nuclear HDAC6 protein expression exhibited significant model effects (*p* < 0.001 for each, Figure 3c,d), and this marker was reduced in both tissue fractions 24 h following bout 1. Cytoplasmic and nuclear LDHA protein expression showed no significant model effects (*p* = 0.183 and *p* = 0.076, Figure 3f,g).

### 2.2. Global Protein Lactylation and Acetylation Markers Remain Unaltered Following 60 min of Cycling

Global protein lactylation and acetylation data for the acute cycling bout can be found in Figure 4. Global protein lactylation and global protein acetylation showed no significant model effects (*p* = 0.173 and *p* = 0.482, respectively, Figure 4a,b).

### 2.3. Lactate Administration Does Not Alter Myotube Diameter, Nuclear, or Cytoplasmic Puromycin-Labeled Proteins in C2C12 Cell Line

C2C12 cell culture experiment data can be found in Figure 5. Cytoplasmic and nuclear protein lactylation levels were not significantly different between treatments (ANOVA *p* = 0.714 and *p* = 0.123, respectively, Figure 5a,b). Cytoplasmic and nuclear protein acetylation levels were also not significantly different between treatments (ANOVA *p* = 0.540 and *p* = 0.732, respectively, Figure 5c,d). Myotube diameters were not significantly different between treatments (ANOVA *p* = 0.566, Figure 5f), and this is supported by cytoplasmic puromycin-labeled proteins (i.e., muscle protein synthesis levels) not being different between treatments (*p* = 0.180, Figure 5g).

## 3. Discussion

Given that the current literature is mixed regarding the ability of exercise to affect skeletal muscle protein lactylation, and that acetylation can compete with lactylation as a post-translational modification, the objective of our study was to investigate how different modalities of exercise (acute resistance exercise, chronic resistance training, and acute endurance training) affected these outcomes. We hypothesized that markers associated with protein lactylation would remain unaltered in response to different exercise paradigms and that markers associated with protein acetylation would be dynamically altered. Additionally, we hypothesized that the administration of physiologically relevant lactate concentrations in vitro would not alter markers associated with protein lactylation. The most intriguing finding from our study is the lack of protein lactylation across all experiments. As mentioned prior, some rodent data indicate that treadmill exercise acutely increases protein lactylation [15]. Additionally, others have reported that lactate administration in C2C12 cells promotes histone lactylation [11]; note that cells were treated with 15 mM sodium lactate for 3–5 days and their model was likely supraphysiological compared to ours, which attempted to emulate lower-dose treatments that are observed during exercise. This notwithstanding, in line with our prior human data [16], the current data add further evidence to suggest that: (i) resistance exercise does not acutely or chronically affect nuclear or cytoplasmic protein lactylation; (ii) one bout of endurance exercise (via cycling) does not acutely affect skeletal muscle protein lactylation; and (iii) treating myotubes with lower, higher, or a high pulsatile “exercise-like” dose of sodium lactate does not affect myotube lactylation.

Nuclear and cytoplasmic HDAC2 and LDHA protein levels were also not acutely or chronically altered with resistance training. Both enzymes have been implicated in catalyzing cellular protein de-lactylation and lactylation by acting as a Kla eraser and writer, respectively [9,17,18]. While not measured in the current study, it is also notable that lactyl-CoA is used as a substrate by Kla writers to lactylate proteins. According to Varner et al. [19], lactyl-CoA in cardiac tissue is exceedingly low relative to other acyl-CoAs, and this too casts doubt as to whether muscle proteins can be dynamically lactylated in response to exercise stressors. It is notable, however, there is recent evidence in humans to suggest that heightened skeletal muscle protein lactylation is associated with metabolic dysregulation. In this regard, Maschari et al. [17] reported that obese, insulin-resistant women presented ~20% higher skeletal muscle protein lactylation levels compared to their lean, healthy counterparts. In explaining their findings, the researchers identified other studies demonstrating increased post-translational modifications to muscle proteins that accompany metabolic dysfunction (e.g., acetylation and malonylation) and speculated that chronic/low-level lactate accumulation via metabolic dysregulation leads to increased skeletal muscle protein lactylation. With this collective evidence in mind, we posit that skeletal muscle protein lactylation is likely not a post-translational protein modification appreciably involved in the adaptive response to training.

The main motivation for measuring skeletal muscle and myotube protein acetylation levels was to determine if an inverse pattern was evident relative to protein lactylation levels across experiments. Additionally, we opted to assess HDAC2 and HDAC6 protein levels with the resistance training study specimens given that (unlike lactylation states) nuclear and cytoplasmic protein acetylation patterns were altered, the former two proteins act to deacetylate cellular proteins, and the latter marker is a histone acetyltransferase enzyme [20]. The observed outcomes are interesting in the context of other human exercise studies. For instance, McGee et al. [5] reported that one bout of cycling acutely reduces nuclear HDAC abundance following exercise. Hostrup et al. [21] more recently used an acetylomic approach to report that five weeks (15 sessions) of high-intensity cycle interval training increased the acetylation of ~260 sites of mitochondrial TCA cycle proteins.

Lim et al. [12] reported that histone 3 acetylation increased three hours following one resistance exercise bout, as well as following 10 weeks of resistance training. Interestingly, we observed that nuclear and cytoplasmic HDAC6 and protein acetylation levels decreased following one bout and chronic resistance training. The reduction in nuclear HDAC6 agrees in principle with the aforementioned report by McGee et al. However, reduced muscle protein acetylation with resistance training contrasts with the notion that exercise training generally increases protein acetylation. While this finding is difficult to reconcile, Hain et al. [22] reported that C2C12 myotube atrophy (induced via serum-conditioned media experiments from a cancerous cell line) coincided with significantly heightened myotube protein acetylation. In explaining their findings, the researchers posited that protein hyperacetylation can reduce protein function and stability, and this is a sign of muscle atrophy. Notably, over 80% of contractile proteins in skeletal muscle are subject to acetylation [23]. Acetylation has also been found to be a master regulator in mitochondrial metabolic pathways, with 63% of mitochondrial proteins having been found containing lysine acetylation sites [24]. Collectively, these results indicate that reductions in protein acetylation could be an explicit response to resistance exercise to potentially aid in the adaptive response. However, this is highly speculative, and future investigations are needed to determine protein- and site-specific acetylation events along with the functional ramifications of reduced protein acetylation.

Lastly, other secondary cell culture findings are noteworthy due to select in vitro and rodent evidence indicating that lactate may potentiate anabolic signaling in skeletal muscle [25]. However, as we have pointed out previously [16], single- or multi-day lactate administration paradigms have not been shown to bolster anabolic signaling in rodent or human skeletal muscle [26,27], and our data continue to support this notion.

### Experimental Considerations

We aimed to leverage tissue collected from prior studies performed by our laboratories to further explore the potential for exercise stimuli to affect skeletal muscle protein lactylation. A major limitation to the current study is the lack of peri-exercise blood draws and/or the assessment of intramuscular lactate concentrations in human studies. However, we are relatively confident that the resistance training protocol, as well as the 60 min cycling bout, elevated both metrics based on prior human data. For instance, we have previously reported that lower-body resistance training increases peri-exercise blood lactate concentrations ~7.2-fold [16] without affecting peri-exercise skeletal muscle protein lactylation. Additionally, as little as 3 min of cycling exercise at ~80% VO2max has been shown to increase blood lactate levels ~6–8-fold in trained and untrained individuals [28]. The current data are also limited in participant number and biopsy sampling time points. Therefore, we cannot completely rule out that resistance or endurance exercise does not affect skeletal muscle protein lactylation beyond the time points assayed herein, and future human studies with more sampling time points are needed. Additionally, although global nuclear/cytoplasmic protein lactylation levels were shown not to be affected across experiments, it remains possible that specific lysine residues across various proteins may exhibit altered lactylation states. Hence, lactylomic-based approaches can serve to further investigate this potential phenomenon. While enzymes that affect protein acetylation and lactylation were examined using Western blotting, we did not assess the relative activities of these enzymes due to lysate and budget constraints. Hence, adding these assays in future work will provide more insight. Finally, given that skeletal muscle protein acetylation was measured to merely examine if an inverse/competitive pattern was present relative to protein lactylation levels, we view our resistance training protein acetylation data as being underdeveloped. Whereas other studies indicate that endurance exercise generally increases skeletal muscle protein acetylation, the observation that resistance training reduced muscle protein acetylation was unexpected and warrants further investigation.

## 4. Materials and Methods

### 4.1. Ethical Approval for Human Work

Human vastus lateralis muscle specimens were obtained from apparently healthy, untrained college-aged participants described in two previously published studies by Chaves et al. [29] (acute and chronic resistance training) and Roberson et al. [30] (acute cycling). The resistance training study by Chaves et al. was carried out at the Federal University of São Carlos (IRB protocol #: 5.505.441). The study was conducted in accordance with the most recent version of the Declaration of Helsinki and was pre-registered as a clinical trial (Brazilian Registry of Clinical Trials—RBR-57v9mrb). The study by Roberson et al. was approved by the Auburn University Institutional Review Board (protocol #18-226) but was not pre-registered as a clinical trial. Inclusion criteria for both studies can be found in the original works.

### 4.2. Study Designs

A schematic illustrating study designs is depicted in Figure 1 above, and descriptions for each experiment follow.

#### 4.2.1. Acute and Chronic Resistance Training Study

Muscle specimens were obtained from 12 participants [8M/4F, 25 ± 4 years old, 24.0 ± 3.8 kg/m^2^), and training as well as specimen collection were performed at the Federal University of Sao Carlos. The resistance training protocol consisted of four sets of 9–12 maximum repetitions of unilateral leg extension exercises, with a 90-s rest period between sets. The load was adjusted for each set to ensure that concentric muscle failure occurred within the target repetition range. Participants completed 24 training sessions over a period of 10 weeks, with sessions conducted 2 to 3 times per week. Four mid-thigh vastus lateralis (VL) biopsies were obtained during the intervention including the basal state pre-intervention sample (pre, first biopsy), 24 h after the first training bout (second biopsy), 96 h after the second to last training bout (basal state post-intervention biopsy, third biopsy), and 24 h after the last training bout (trained state acute response, fourth biopsy). However, due to sample limitations, we only analyzed the first three biopsies with the participants reported herein. Tissue was lysed using a commercially available nuclear isolation kit (Abcam; Cambridge, MA, USA; Cat. No. ab113474) as previously described [16], and nuclear as well as cytoplasmic lysates were prepared for Western blotting as described in the next section.

#### 4.2.2. Acute Endurance Training Study

Nine participants (3M/6F, 23 ± 2 years old, 23.1 ± 2.6 kg/m^2^) reported to the laboratory during the morning hours under fasted conditions and donated a baseline VL biopsy. Participants then mounted a cycle ergometer (Velotron; RacerMate, Seattle, WA, USA) and performed a 5 min warm-up at a self-selected pace. Wattage was adjusted thereafter to achieve 70% heart rate reserve, and participants cycled for 60 min. Participants exhibited an average heart rate of 161 ± 9 beats/minute (resting heart rate average = 74 ± 6 beats per minute). Post-exercise biopsies were then obtained 2 and 8 h following the cycling bout. Whole-tissue lysates were obtained using a general cell lysis buffer (Cell Signaling; Danvers, MA, USA; Cat. No. 9803), as previously described by Roberson et al. [30] and processed for Western blotting, as described below.

#### 4.2.3. Cell Culture Experiments

C2C12 myoblasts (passage 2–4, ATCC; Manassas, VA, USA; Cat. No. CRL-1772) were cultured in 100 mm plates with DMEM (Corning; Corning, NY, USA) supplemented with 10% fetal bovine serum (VWR; Radnor, PA, USA; Cat No. 89510-182), 1% penicillin/streptomycin (VWR; Cat No. 16777-164), and 0.1% gentamicin (VWR; Cat No. 97061-372) at 37 °C in a humidified atmosphere of 5% CO_2_. Upon reaching 90% confluency, myoblasts were cultured in differentiation media (DM) containing DMEM medium supplemented with 2% horse serum (Corning; Cat No. 35-030-CV), 1% penicillin/streptomycin, and 0.1% gentamicin for 7 days until mature myotube formation. To investigate the influence of lactate on C2C12 myotube protein lactylation and anabolism, myotubes (3–4 replicates per condition) were treated with varying levels of sodium lactate spiked into DM (LOW: 1 mM for 24 h, PULSE: 10 mM for 30 min followed by 1 mM for 23.5-h, HIGH: 10 mM for 24 h) (ThermoScientific; Waltham, MA, USA; Cat. No. L14500). Twenty-three and one-half hours into treatments cells were pulse-labeled with 1 µM of puromycin hydrochloride (VWR; Cat. No. 97064-280) for the assessment of cytoplasmic muscle protein synthesis levels using the SUnSET method as performed with in vitro studies in our laboratory [31,32]. Cells were then lysed using a commercially available nuclear isolation kit (Abcam; Cat. No. ab113474), and nuclear as well as cytoplasmic lysates were prepared for Western blotting as described in the next section.

A second set of myotube plates with each sodium lactate treatment was also immuno-stained to assess morphology. Following the treatments described above, myotubes were fixed with 10% formalin (VWR) for 15 min at room temperature, then washed 3 × 3 min with PBS containing 0.2% Triton X-100 (PBS/Triton). Cells were then blocked with PBS/Triton containing 1% BSA for 1 h at room temperature, followed by incubation with a primary antibody solution containing anti-myosin heavy chain (1:100) (DSHB; Iowa City, IA, USA; Cat. No. A4.1025) in PBS/Triton/BSA for 3 h at room temperature. Following 3 × 3 min washes with PBS/Triton, cells were incubated with a secondary antibody solution containing goat anti-mouse IgG2a AF488 (ThermoFisher; Cat. No. A-21131) in PBS/0.2% Triton-X for 2 h at room temperature. Cells were then washed 3 × 3 min with PBS/Triton and incubated with DAPI (ThermoFisher; Cat. No. D3571) for 10 min. After the last wash, multiple images were obtained by a fluorescent microscope using a 10× objective (Zeiss Axio imager.M2; Carl Zeiss Microscopy; Jena, Germany). Myotube diameters were then quantified on digital images using ImageJ v1.52a (National Institutes of Health; Bethesda, MD, USA), like methods we have previously described [32].

### 4.3. Western Blotting

Nuclear and cytoplasmic isolates (resistance training and C2C12 experiments) as well as general lysates (acute cycling) were assayed for total protein content using a commercially available BCA Protein Assay Kit (ThermoFisher; Waltham, MA, USA) and a spectrophotometer (Agilent Biotek Synergy H1 hybrid reader; Agilent, Santa Clara, CA, USA). Thereafter, protein concentrations were standardized to 0.5–1.0 µg/µL using Laemmli buffer depending upon the protein fraction, and 15 μL of each sample was loaded into 4–15% SDS-polyacrylamide gels (Bio-Rad; Hercules, CA, USA). Electrophoresis occurred at 180 V for 50 min using 1× SDS-PAGE run buffer (VWR). Following SDS-PAGE, proteins were transferred to polyvinylidene difluoride membranes (Bio-Rad) at 200 mA for 2 h. Following transfers, membranes were Ponceau stained and imaged using a gel documentation system (ChemiDoc Touch; Bio-Rad) to capture whole-lane images for protein normalization purposes. Membranes were then blocked for 1 h at room temperature with 5% nonfat milk powder mixed in Tris-buffered saline solution containing 0.1% Tween-20 (TBST; VWR).

For the interrogation of protein lactylation, membranes were incubated for 48 h at 4 °C with a rabbit anti-lactyl lysine antibody (ThermoFisher Scientific; Waltham, MA, USA; Cat. No. PA5116901) at a 1:1000 dilution in TBST solution containing 5% bovine serum albumin (BSA). Following primary antibody incubations, membranes were washed three times in TBST (15 min total) and incubated with a horseradish peroxidase-conjugated anti-rabbit antibody (1:2000; Cell Signaling; Cat. No. 7074) in TBST solution containing 5% BSA at room temperature for 1 h. After secondary antibody incubations, membranes were washed three times in TBST (15 min total) and developed in a gel documentation system (ChemiDoc Touch; Bio-Rad) using a chemiluminescent reagent (Luminata Forte HRP substrate; Millipore Sigma; Burlington, MA, USA). Software indicated that none of the bands were overexposed and presented grayscale values that were not above the maximal detectable signal.

Other membranes were incubated overnight at 4 °C with either an anti-acetyl lysine antibody (Cell Signaling; Cat. No. 9441), anti-HDAC2 antibody (Cell Signaling; Cat. No. 57156), anti-HDAC6 antibody (Cell Signaling; Cat. No. 7558), anti-LDHA antibody (Cell Signaling; Cat. No. 2012), anti-p300 antibody (Cell Signaling; Cat. No. 86377), or anti-acetylated-histone 3 (H3K9) antibody (Cell Signaling; Cat. No. 8173) at 1:1000 dilutions in TBST solution containing 5% BSA. Following primary antibody incubations and three TBST washes, membranes were incubated with horseradish peroxidase-conjugated anti-rabbit antibody (1:2000) in TBST solution containing 5% BSA at room temperature for 1 h before development, as described above. Notably, the p300 immunoblotting experiments did not yield any signal suggestive of protein expression. Upon reviewing some of our previously published deep proteomic data on skeletal muscle in college-aged and older adults, the lack of signal herein may potentially be due to its low level of expression in adult skeletal muscle, as revealed by the lack of detection (note, this counters other nuclear proteins such as histones, which are highly abundant) [33].

Lastly, our cell culture lysates underwent the same anti-acetyl lysine antibody and anti-lactyl lysine antibody protocols described above, and membranes were also interrogated using an anti-puromycin antibody (Sigma-Aldrich, St. Louis, MO, USA, Cat. No. MABE342) at 1:10,000 dilution in TBST solution with 5% BSA. Following primary antibody incubations and three TBST washes, membranes were incubated with horseradish peroxidase-conjugated an anti-mouse antibody (1:2000; Cell Signaling; Cat. No. 7072) at a 1:2000 dilution in TBST solution containing 5% BSA at room temperature for 1 h before development, as described above.

Whole lane or band densities from all Western blot experiments were analyzed using ImageLab v6.0.1 (Bio-Rad). Protein target values were corrected for Ponceau densities and normalized to pre values (exercise studies) or the LOW treatment (C2C12 experiments) to obtain data as fold-change values.

### 4.4. Statistics

All statistical analyses were performed using GraphPad Prism v9.2.0 (San Diego, CA, USA), and the figures were created using either GraphPad Prism or BioRender (https://biorender.com, accessed 14 August 2024). For the exercise studies, one-way repeated-measures ANOVAs were performed, and Tukey’s post hoc tests were conducted if model significance was obtained. For the cell culture study, ordinary one-way ANOVAs were performed. All data in the text and in figures are presented as mean ± standard deviation (SD) values, and statistical significance was established as *p* < 0.05.

## 5. Conclusions

This is the second human study to date demonstrating that different exercise stimuli do not affect global skeletal muscle protein lactylation, and these data continue to challenge lactate as being a signaling metabolite that affects muscle anabolism. Moreover, the resistance exercise-induced decrement in global skeletal muscle protein acetylation levels is a relatively novel finding, and the significance of this finding requires additional research.

## Figures and Tables

**Figure 1 ijms-25-12216-f001:**
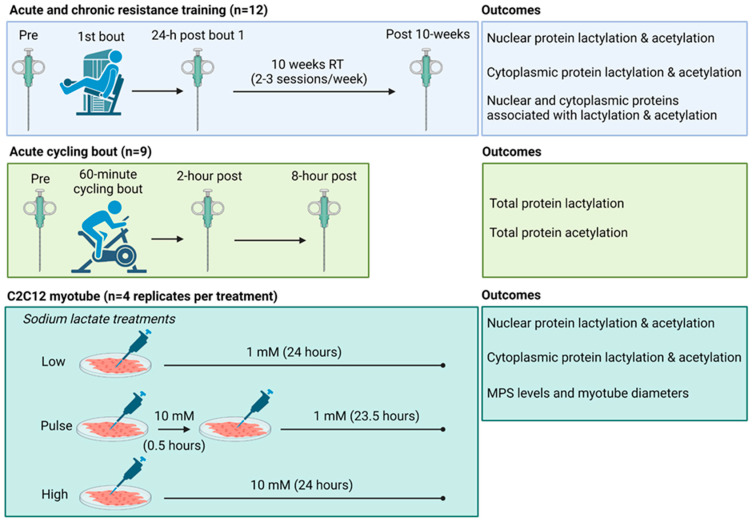
Study schematic. This graphic gives a visual overview of the acute and chronic resistance training study, the acute cycling study, and cell culture model (constructed using Biorender.com, accessed on 14 August 2024).

**Figure 2 ijms-25-12216-f002:**
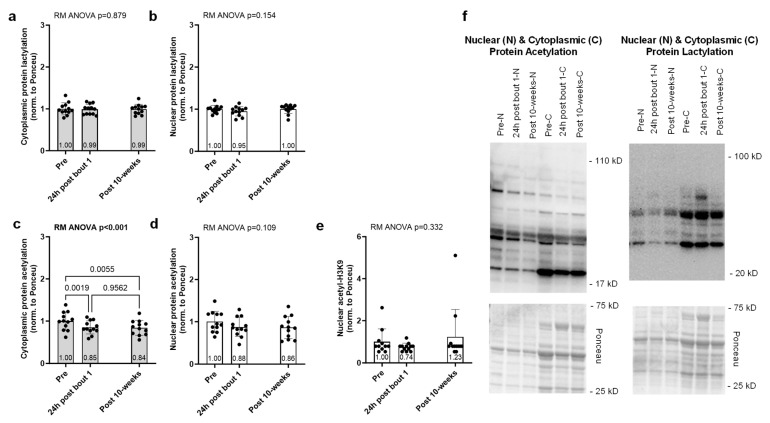
Effects of acute and chronic resistance training on nuclear and cytoplasmic protein lactylation and acetylation. Biopsies were obtained prior to the intervention (pre), 24 h following the first exercise bout (24 h post-bout 1), and following the 10-week intervention (post-10 weeks) in 12 participants. No significant alterations occurred with cytoplasmic protein lactylation (**a**), nuclear protein lactylation (**b**), nuclear protein acetylation (**d**), or nuclear H3K9 acetylation (**e**). However, cytoplasmic protein acetylation (**c**) was significantly reduced with one bout of resistance exercise and chronic training. Representative Western blots for the protein targets (**f**) are presented. Bar graphs are mean and standard deviation values, with individual respondent data overlaid.

**Figure 3 ijms-25-12216-f003:**
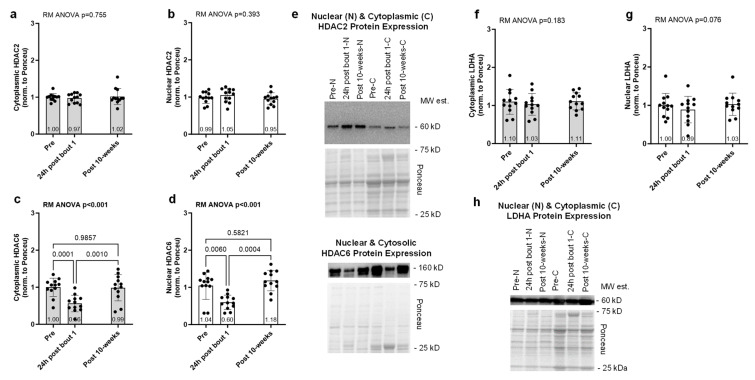
Acute and chronic resistance effects on the expression of select nuclear and cytoplasmic proteins associated with acetylation and lactylation. Biopsies were obtained prior to the intervention (pre), 24 h following the first exercise bout (24 h post-bout 1) and following the 10-week intervention (post-10 weeks) in 12 participants. No significant alterations occurred with cytoplasmic or nuclear HDAC2 protein expression (**a**,**b**). Cytoplasmic and nuclear HDAC6 protein levels were significantly reduced with one bout of resistance exercise (**c**,**d**). No significant alterations occurred with cytoplasmic or nuclear LDHA protein expression (**f**,**g**). Representative Western blots for the protein targets (**e**,**h**) are presented. Bar graphs are mean and standard deviation values with individual respondent data overlaid.

**Figure 4 ijms-25-12216-f004:**
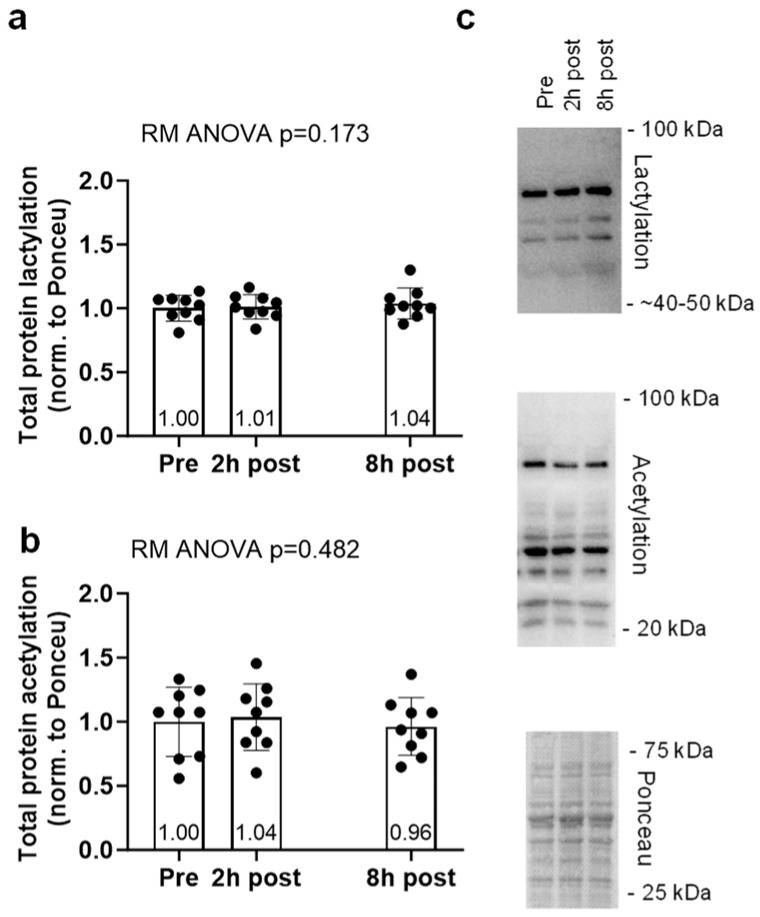
Acute cycling bout data. Global protein lactylation (**a**) and global protein acetylation (**b**) markers assessed at baseline, 2 h post-, and 8 h post-60 min cycling bout. Representative Western blots for the protein targets (**c**) are presented. Bar graphs are mean and standard deviation values, with individual respondent data overlaid.

**Figure 5 ijms-25-12216-f005:**
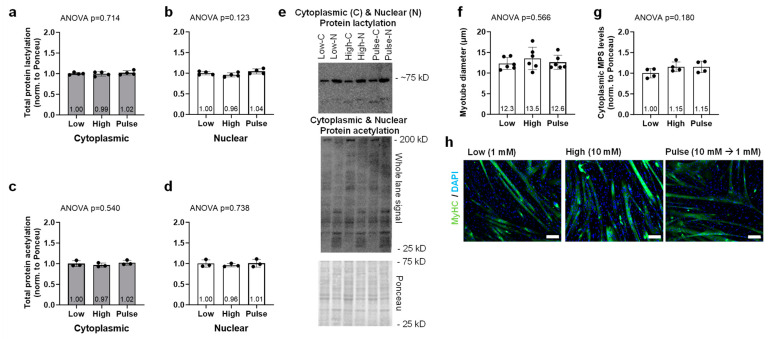
C2C12 sodium lactate treatment data. Cytoplasmic (**a**) and nuclear (**b**) protein lactylation levels were not different between treatments. Cytoplasmic (**c**) and nuclear (**d**) protein acetylation levels were also not different between treatments. Finally, myotube diameters (**f**) and cytoplasmic MPS levels (**g**) were not significantly different between treatments. Representative Western blots for the protein targets (**e**) are presented, and representative myotube images are also presented (**h**); note inset white bars are 100 µm. Bar graphs are mean and standard deviation values with individual replicate values overlaid.

## Data Availability

The data that support the findings of this study are available from the corresponding author (mdr0024@auburn.edu) upon reasonable request.

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
