# Peer review of "Acute and Chronic Resistance Training, Acute Endurance Exercise, nor Physiologically Plausible Lactate In Vitro Affect Skeletal Muscle Lactylation"

_ijms, 2024, doi:10.3390/ijms252212216_

Round 1

Reviewer 1 Report

Comments and Suggestions for Authors

In the current study, the authors tested the role of chronic resistance training and/or one bout of endurance exercise on skeletal muscle protein acetylation and lactylation as well as the effect of sodium lactate treatments on C2C12 myotubes. This is an interesting and well-written study.

Nevertheless, the representative blots do not correlate with some of the data presented in the graphs (Figure 2 and Figure 4). This requires explanations/clarifications. Based on the blots, there are increases in protein lactylation at ~50kDa and protein acetylation at ~50kDa and ~80kDa in Figure 2. Similarly, there is an increase in protein lactylation at ~80kDa in Figure 4. Maybe regions with major bands should be quantified and compared separately to identify the differences present in the blots.

Suggestions:

Please add labels to the Original Images. It is impossible to understand what the lanes represent.

Lines 76-82: It is stated that there are no differences in the nuclear protein lactilation. On the blot in Figure 2F there is a clear difference between Pre-N and Post 10-weeks-N. Ponceau staining does not have this difference. Similarly, there is a clear difference on the blot in Figure 2F between Pre-N and Post 10-weeks-N acetylation and it is an increase. Please explain these differences. Maybe your blot quantification method was not very sensitive.

Figure 3e and h. All blots presented in these figures are extremely overexposed. It is impossible to obtain reliable quantifications by using these blots.

On the blot in Figure 4c, there is a clear difference between Pre- and Post-8h lactylation, and it does not reflect the data in Figure 4a.

The blots in Figure 5e are overexposed. Please provide a better image.

Lines 239-244: Please provide protocol numbers and dates of approval for all human experiments.

Line 276: Where the C2C12 cells were obtained from?

Line 281: Was sodium lactate added in the presence of 2% horse serum? Could this have an effect on the experiment by lactylation of serum proteins and decreasing the concentration of sodium lactate in the media?

Line 336: Which figure has anti-p300 antibody data?

Author Response

Thank you kindly for the reviews.  Please see PDF attachment.

Reviewer 2 Report

Comments and Suggestions for Authors

The manuscript ijms-3266790 is interesting and has the ambitious goal of shedding light on the regulations induced in skeletal muscle after exercise in terms of lactylation and acetylation. These two regulations are interesting new post-translational regulations that compete for the regulation of skeletal muscle activity.

Some aspects need to be clarified before accepting the manuscript for publication in IJMS.

Major revision.

1) Even if they are already known, the authors should clearly indicate the circulating levels of lactic acid during the experimental protocols.

Possibly also the intracellular values ​​would be useful.

2) As for the levels of HDAC2, HDAC6 and LDHA, it would be useful to have data regarding the enzymatic activity in addition to the protein levels.

3) The authors have several valuable biological material at their disposal. It would be useful to have the levels of other proteins such as MHC, (even better if the different isoforms) or other muscle-specific proteins to see if there is a modulation induced by the alteration (presumed or possible) of lactate levels.

Author Response

(The authors gave the same response as above.)

Round 2

Reviewer 1 Report

Comments and Suggestions for Authors

The authors addressed most of my critiques. I highly recommend replacing the Western Blot image for HDAC2 in Figure 3e with a better image. The background in the current image is extremely dark and it is difficult to compare specific bands. The original image for HDAC2 that you provided has a much better specific band-to-background ratio.

Author Response

We have replaced the HDAC2 image in Fig 3e.  Thank you for all your help.  We believe that your comments improved the paper.

Reviewer 2 Report

Comments and Suggestions for Authors

In the previous review, three issues were highlighted:

1) Lack of lactate test. I wonder why a lactacimeter was not used in which one can analyze lactate levels with a drop of blood from an earlobe.
The authors responded to this point by citing themselves and referring to work already done. The limitation was correctly inserted in the manuscript.

2) Enzyme activities. On this point, I understand the economic-practical limitations but the authors agree with me that it is a limitation.
This limitation should be inserted in the manuscript.

3) Greater analysis of muscle tissue. As in the previous point, I understand the economic-practical limitations, however a small biopsy sample would be sufficient to perform several analyses on MHC isoforms.
In any case, I would indicate in the discussion that further studies should be carried out regarding the modulation of muscle-specific proteins induced by lactic acidification and acetylation.

Author Response

We will address #2 and 3 below given that you've indicated #1 was sufficiently addressed.

For number 2, we have inserted the following into lines 232-235: "While enzymes that affect protein acetylation and lactylation were examined using western blotting, we did not assess the relative activities of these enzymes due to lysate and budget constraints. Hence, adding these assays in future work will provide more insight."

For number 3, we had the following in our prior draft: "Additionally, although global nuclear/cytoplasmic protein lactylation levels were shown not to be affected across experiments, it remains possible that specific lysine residues across various proteins may exhibit altered lactylation states. Hence, lactylomic-based approaches can serve to further interrogate this potential phenomenon."

However, to better highlight this point, we have also inserted the following in the abstract:

"However, further human research with more sampling timepoints and a lactylomics approach are needed to determine if, at all, different exercise modalities affect skeletal muscle protein lactylation."